# Association of Age-Related Macular Degeneration with Prior Hyperthyroidism and Hypothyroidism: A Case–Control Study

**DOI:** 10.3390/jpm12040602

**Published:** 2022-04-08

**Authors:** Shih-Han Hung, Sudha Xirasagar, Tung-Mei Tammy Kuang, Wei-Wen Chang, Yen-Fu Cheng, Nai-Wen Kuo, Herng-Ching Lin

**Affiliations:** 1Department of Otolaryngology, School of Medicine, College of Medicine, Taipei Medical University, Taipei 110, Taiwan; seedturtle@gmail.com; 2Department of Otolaryngology, Wan Fang Hospital, Taipei Medical University, Taipei 110, Taiwan; 3International Master/Ph.D. Program in Medicine, College of Medicine, Taipei Medical University, Taipei 110, Taiwan; 4Department of Health Services Policy and Management, Arnold School of Public Health, University of South Carolina, Columbia, SC 29208, USA; sxirasagar@sc.edu; 5Department of Ophthalmology, Taipei Veterans General Hospital, Taipei 112, Taiwan; kuangtammy@gmail.com; 6Department of Ophthalmology, School of Medicine, National Yang Ming Chiao Tung University, Taipei 112, Taiwan; 7Division of General Surgery, Department of Surgery, Wan Fang Hospital, Taipei Medical University, Taipei 110, Taiwan; weiwenabow@gmail.com; 8Department of Medical Research, Taipei Veterans General Hospital, Taipei 112, Taiwan; yfcheng2@vghtpe.gov.tw; 9Department of Otolaryngology-Head and Neck Surgery, Taipei Veterans General Hospital, Taipei 112, Taiwan; 10Faculty of Medicine, National Chiao Tung University, Taipei 112, Taiwan; 11School of Health Care Administration, College of Management, Taipei Medical University, Taipei 110, Taiwan; nwkuo@tmu.edu.tw; 12Sleep Research Center, Taipei Medical University Hospital, Taipei 110, Taiwan

**Keywords:** age-related macular degeneration, hypothyroidism, hyperthyroidism, epidemiology

## Abstract

Prior studies suggest a possible association between thyroid disease and the subsequent development of age-related macular degeneration (AMD), although it remains inconclusive. This study aimed to evaluate the association of AMD with prior hyper-/hypothyroidism based on nationwide population-based data. We retrieved records of the study patients from the National Health Insurance Research Database, 7522 patients with a first-time diagnosis of AMD and 7522 propensity score-matched controls. Multiple logistic regression analyses were performed to explore the association of neovascular AMD with previously diagnosed hyperthyroidism or hypothyroidism. The Chi-square test shows that there was a statistically significant difference in the prevalence of prior hyperthyroidism between cases and controls (1.18 vs. 0.13%, *p* < 0.001). Furthermore, there was a statistically significant difference the prevalence of prior hypothyroidism between cases and controls (0.44 vs. 0.69%, *p* < 0.001). Multiple logistic regression analysis reveals that AMD was statistically and significantly associated with prior hyperthyroidism after adjusting for age, sex, monthly income, geographical location, urbanization level, hypertension, hyperlipidemia, diabetes, and coronary heart disease (odds ratio (OR) = 9.074, 95% CI = 4.713–17.471). The adjusted OR of prior hypothyroidism in patients with AMD was 3.794 (95% CI: 2.099~6.858) when compared to the controls. We conclude that patients with thyroid dysfunction are at higher risk of developing AMD Results suggest that these patients could benefit from proactive regular eye checkups to detect evolving eye pathology, even while vision remains normal during the initial phases.

## 1. Introduction

Age-related macular degeneration (AMD) is a degenerative disease of the central portion of the retina that results primarily in loss of central vision. With population aging, AMD is becoming a major public health problem with serious disability implications as central vision is required for daily activities such as walking, driving, reading, and performing the activities of daily living [1]. It has been estimated that as aging populations are significantly growing in many countries, more than 20% of the older population may have the disorder. In the United States, age-related macular degeneration was estimated to affect more than 1.75 million individuals in 2004, rising to almost 3 million by 2020 due to rapid growth in the aging population [2]. The disease usually affects people over 60 years old. While many patients with macular involvement may maintain daily life activities, such as walking without physical aids, using their peripheral vision alone, their quality of life is substantially eroded due to much impaired central vision.

AMD is described as a progressive condition that evolves through phases, classified as early-stage (medium-sized drusen and retinal pigmentary changes) and late-stage (neovascular and atrophic). Early-stage disease may develop gradually and asymptomatically over several years to advanced AMD. Three phases, each associated with a distinct risk level for developing vision loss are recognized: 1. Early AMD, characterized by multiple small drusen (<63 μm) or intermediate drusen (≥63 μm and <125 μm) with no evidence of advanced AMD features; 2. Intermediate AMD, characterized by extensive intermediate drusen or large drusen (≥125 μm) without features of advanced AMD; 3. Advanced AMD, characterized by the presence of one or the other feature, namely geographic atrophy or neovascular age-related macular degeneration [3].

Currently, the disease has been regarded as having multifactorial origins, influenced by a combination of environmental factors and specific genetic variants associated with dysregulation of the complement, lipid, angiogenic, inflammatory, and extracellular matrix pathways [4]. More than 50 genetic susceptibility loci have been identified, including the most important loci in the CFH and ARMS2 genes [5]. Interestingly, the association between AMD and thyroid diseases is little addressed.

A population-based cohort study by Chaker et al. showed that higher FT4 values are associated with increased risk of AMD—even in euthyroid individuals. The authors suggested that the thyroid hormone plays an important role in pathways leading to AMD [6]. Further, case studies evaluating serum thyroid-stimulating hormone (TSH) and free thyroxine (FT4) measurements and thyroid dysfunction (hyperthyroidism and hypothyroidism) suggest that overt hyperthyroidism is independently associated with an increased risk of incident AMD. Thyroxine usage in older adults was also positively associated with the incidence of AMD [7]. Thyroid cancer patients older than 50 years old were found to have an increased risk of AMD in a longitudinal data-based study [8].

This observation is not surprising, because the lack of thyroid hormone is associated with cone photoreceptor preservation in animal models. However, the observed beneficial effect of an absence of thyroid hormone on AMD is not consistent [9,10]. Bromfield et al. reported a possible relationship between AMD and hypothyroidism [11]. In a recent meta-analysis regarding the association between thyroid disease and AMD risk, the authors concluded that thyroid disease is associated with higher AMD risk, and thyroid disease prevention strategies may have a significant effect on the prevention of AMD and warrant further evaluation [12].

However, although some prior studies support a possible association of thyroid disease with AMD risk, evidence regarding this association remains inconclusive [12]. Almost all studies have used clinical data from regional databases or data from selected hospitals or sub-populations of patients. Their findings may be biased from misclassification bias for either the causal variable or the outcome, due to healthcare use at facilities/regions outside the scope covered by the database used. The purpose of this study was to evaluate the association of AMD with prior hyper-/hypothyroidism based on a nationwide population-based database.

## 2. Methods

### 2.1. Database

We conducted a nationwide population-based case-control study in Taiwan. We identified study patients from the National Health Insurance Research Database (NHIRD). Taiwan implemented the National Health Insurance (NHI) program from 1995 to provide affordable health care to all of the country’s citizens. The NHI program is a mandatory social insurance program covering all Taiwanese citizens. As of 2021, 99% of Taiwan’s population were enrolled. The Bureau of National Health Insurance (BNHI) receives claims from all professional and institutional providers of health care in Taiwan and provides sorted claims data by provider type to the NHIRD. The NHIRD includes registration files of NHI beneficiaries and files with claims for ambulatory care visits, details of ambulatory care orders, inpatient expenditures by admissions, details of inpatient orders, and details of prescriptions dispensed at contracted pharmacies. Many scientists in Taiwan have used the NHIRD databases to conduct epidemiological studies of diseases and treatments using claims data on follow-up medical services. 

The study was approved by the institutional review board of Taipei Medical University (TMU-JIRB N202202020) and is compliant with the Declaration of Helsinki. Because we used deidentified administrative data, informed consent was waived.

### 2.2. Identification of Cases and Controls

For case selection, we first identified all patients with a first-time diagnosis of AMD (ICD-9-CM code 362.50, 362.51, 362.52, or ICD-10-CM code H35.31 or H35.32) during an ambulatory care visit to clinics or outpatient departments of hospitals between 1 January 2015 and 31 December 2016, to a toral of 7902 patients. Because of concerns about diagnostic validity in administrative datasets, we selected patients with an AMD diagnosis in at least two claims filed by board-certified ophthalmologists between 1 January 2015 and December 31, 2016 (*n* = 7738). We defined the date of first diagnosis of AMD as the index date for cases. We excluded 216 patients aged under 40 years due to the very low incidence of AMD in this age group. The remaining 7522 patients with AMD were included as cases.

We retrieved matched controls from the remaining NHIRD enrollees ≥40 years old from the Registry of Beneficiaries. We selected controls, one per case, using propensity score matching. We first excluded all enrollees who had ever received a diagnosis of macular degeneration. Matching was carried out using patient demographic variables (age, sex, monthly income category, geographic location (Northern, Central, Southern, and Eastern), and urbanization level of the patient’s residence (5 levels, with 1 indicating the most urbanized and 5 the least urbanized)), and medical comorbidities including hyperlipidemia (ICD-9-CM code 272 or ICD-10-CM code E78), diabetes (ICD-9-CM code 250 or ICD-10-CM codes E10~E14), coronary heart disease (ICD-9-CM codes 410–414 or 429.2 or ICD-10-CM codes I20-I25), and hypertension (ICD-9-CM codes 402–404 or ICD-10-CM codes I10~I15). We then entered these variables into a multivariable logistic regression model as predictors to calculate the expected probability that each enrollee with certain characteristics would be assigned to the case group (as opposed to the control group). However, after calculating a propensity score for all enrollees, a score-matched control may not be found for each case. We therefore used the alternative method of nearest neighbor within calipers to match controls (a priori value for the calipers is +/−0.01). The matching ratio can be one-to-one. Furthermore, we matched one control per case based on index year. The selected controls were matched to a given case if they had at least one ambulatory care visit in the year of the index date of case. The date of the control patient’s first ambulatory care visit during the index year of their matched case was defined as their index date. In order to reduce the possibility of misclassification bias due to systematic differences in eye care utilization between cases and controls, we also ensured that the selected controls had received an eye examination by an ophthalmologist within 2 years before the index date. Ultimately, the final study sample included 7522 cases and 7522 controls.

### 2.3. Ascertainment of Exposure 

This study attempted to estimate the odds of having received a diagnosis of hyperthyroidism or hypothyroidism prior to the index date for cases relative to controls. Patients with prior hyperthyroidism or hypothyroidism were identified based on at least two medical care claims within 3 years before the index date showing these diagnoses (ICD-9-CM code 242 (Thyrotoxicosis with or without goiter), 244 (Acquired hypothyroidism), or ICD-10-CM code E05 (Thyrotoxicosis) or E03.9 (Hypothyroidism, unspecified)).

### 2.4. Statistical Analysis

We used the SAS system (SAS System for Windows, V. 9.4, SAS Institute, Cary, NC, USA) for statistical analyses. Descriptive statistics were performed to summarize the characteristics of the sampled patients using mean, standard deviation, count, and percentage for cases and controls. We used Chi-square tests and t-tests to examine differences in demographics (age, sex, monthly income category, geographic location, and urbanization level of the patient’s residence) and medical comorbidities (hyperlipidemia, diabetes, coronary heart disease, and hypertension) between cases and controls. In addition, we carried out a multiple logistic regression analysis to explore the association of neovascular AMD with previously diagnosed hyperthyroidism or hypothyroidism after adjusting for the demographic and co-morbidity variables mentioned above. Two-sided *p* ≤ 0.05 was used to assess statistical significance.

## 3. Results

The sociodemographic characteristics and medical comorbidities of the study patients, 7522 cases and 7522 propensity score-matched controls are shown in Table 1. Chi-square tests and t-tests show no significant differences between cases and controls on age (mean age 68.3 vs. 68.3, *p* = 0.714), sex (male proportion 59.4 vs. 59.2%, *p* = 0.791), monthly income (*p* = 0.292), geographic region (*p* = 0.921), or urbanization level (*p* = 0.764). Among the total 146,308 sampled patients, 64.5% were males. In addition, we found no significant difference in the prevalence of diabetes (28.0 vs. 28.1% *p* = 0.885), hypertension (55.0 vs. 53.9%, *p* = 0.174), coronary heart disease (18.8 vs. 19.1% *p* = 0.724), or hyperlipidemia (38.2 vs. 37.7%, *p* = 0.480) between cases and controls. These findings validate the appropriateness of the propensity score-matching process. 

Table 2 presents the prevalence of prior hyperthyroidism or hypothyroidism among cases and controls. The Chi-square test shows that there was a statistically significant difference in the prevalence of prior hyperthyroidism between cases and controls (1.18 vs. 0.13%, *p* < 0.001). Correspondingly, univariable logistic regression analysis revealed an odds ratio (OR) for prior hyperthyroidism of 8.995 (95% CI: 4.675~17.307) among cases relative to controls. Furthermore, the Chi-square test shows a statistically significant difference in the prevalence of prior hypothyroidism between cases and controls (0.44 vs. 0.69%, *p* < 0.001). Univariable logistic regression analysis shows that the OR for prior hypothyroidism was 3.733 (95% CI: 2.067~6.741) among cases relative to controls.

The results of multivariable logistic regression analysis are presented in Table 3. We found that AMD was significantly associated with prior hyperthyroidism after adjusting for age, sex, monthly income, geographical location, urbanization level, hypertension, hyperlipidemia, diabetes, and coronary heart disease (OR = 9.074, 95% CI = 4.713–17.471).

In addition, multiple logistic regression analysis showed that the adjusted OR of prior hypothyroidism in patients with AMD was 3.794 (95% CI: 2.099~6.858) relative to controls after adjusting for age, sex, monthly income, geographical location, urbanization level, hypertension, hyperlipidemia, diabetes, and coronary heart disease (Table 4).

## 4. Discussion

Our study shows that individuals with prior hyperthyroidism or hypothyroidism have a higher chance of developing age-related macular degeneration later in their life. Our findings of eight-fold increased risk of AMD among patients with hyperthyroidism, and three-fold risk among patients with hypothyroidism, support the importance of euthyroid status in maintaining the health of the retina. This is the first study that used the same population cohort to identify persons with both hyper- and hypothyroid function and evaluated the relationship of each condition to the subsequent development of AMD. 

Risk factors related to AMD have been investigated intensively in the previous literature. Age is the most well-documented risk factor for AMD, addressed in many population-based studies [13,14,15]. It is also known that the female sex is a predisposing factor among the population aged over 75 years [16]. Lifestyle risk factors such as cigarette smoking, which is believed to result in oxidative insults to the retina, is also a documented risk factor with evidence showing a dose–response relationship, the risk of developing advanced AMD increasing with the number of cigarettes smoked [17]. Chronic conditions such as hypertension and high serum cholesterol are also documented risk factors for AMD [18,19].

Racial and genetic factors are also widely studied. AMD is more common among Whites. A recent study showed prevalence rates of AMD among the 45–85 age group to be 2.4% among African-Americans, 4.2% among Hispanics, 4.6% in Chinese Americans, and 5.4% among Caucasians [20]. Specific gene mutations associated with AMD are reported, such as the gene for complement factor H located on chromosome 1q32, and the locus LOC387715(ARMS2)/HtrA1 (high-temperature requirement factor A-1), located on chromosome 10q26 [21,22].

Very little has been documented regarding thyroid function and the pathogenesis of AMD. It was not until 2012 that Bromfield et al. proposed an association between hypothyroidism and AMD. The authors enrolled 9677 patients with hypothyroidism older than 50 years old, of whom 356 participants reported having AMD [11]. As no other details or characteristics were explored, the study limited its finding to the prevalence of the phenomenon and called for further investigation. It was not until animal studies were done that a possible mechanism of development of AMD in relation to thyroid hormone functioning was postulated. Ma et al. used mice with inherited retinal degeneration to evaluate the role of thyroid hormone signaling, which is essential for cell proliferation, differentiation, and apoptosis of retinal cone cells. They found that suppressing thyroid hormone signaling in retinal dystrophy mouse models was protective for cones, providing insights into cone preservation and potential therapeutic interventions [9]. More recently, the group further reported that the inhibition of thyroid hormone signaling protects the retinal pigment epithelium and photoreceptors from cell death in a mouse model investigation of AMD [10]. These studies report findings that differ from those of Bromfield, showing patients with hypothyroidism having higher risk of developing AMD. Many later studies have shown results compatible with findings from human population-based cohorts. In a study from Rotterdam, the researchers studied 5573 participants with a median follow-up of 6.9 years, during which time, 805 people developed AMD. They concluded that higher FT4 values are associated with increased AMD risk, even in euthyroid individuals, as well as increased risk of RPA [6]. A more detailed study by Gopinath et al. further examined the issue by assessing the prospective associations of serum thyroid-stimulating hormone (TSH) and free thyroxine (FT4) levels with thyroid dysfunction (hyperthyroidism and hypothyroidism) and with AMD in 906 participants, making the diagnosis of AMD based on retinal photographs. In this study, participants with overt hyperthyroidism when compared to those with normal thyroid function at baseline had an increased risk of subsequently developing AMD, after adjusting for age, sex, smoking, fish consumption, and variants in AMD susceptibility genes (CFH and ARMS2) with an odds ratio (OR) of 3.51 [7]. The study is particularly illuminating in that those who reported current use of thyroxine versus those who were not current users had a 68% higher risk of incident AMD. Similarly, participants who had ever been on thyroxine medication compared to those who had never been on thyroxine also had a higher risk of any AMD. This result suggests that not only the presence of hyperthyroidism but also the use of thyroxine medication, which is common in patients with hypothyroidism, may have contributed to the development of AMD.

A contradicting finding was reported in a recent meta-analysis. The authors confirmed a significant positive association between thyroid disease and AMD, with an overall relative risk (RR) of 1.25 (95% CI: 1.02, 1.54), but could not find a statistically significant association between thyroid medication and AMD risk. These studies suggest that the relationship between thyroxine and AMD is complicated and warrants further evaluation [12]. 

Several mechanisms may explain the link between thyroid function and AMD. Alfadda et al. studied 10 patients with newly diagnosed overt hypothyroidism and found that after thyroid hormone replacement, ten proteins were more abundant in the hypothyroid vs. euthyroid state, including the complement C3 and C4-A, known to be important in retinal inflammatory and degenerative diseases [23,24,25]. Another possible explanation is through the influence of diabetes mellitus; as patients with diabetes have a higher incidence of thyroid disease than people without, the coexistence of diabetes with thyroid disease leads to retinal endothelial damage. The degree of retinal endothelial cell dysfunctions will significantly impact the course of macro-and microangiopathic complications in these retinopathy patients [26]. Returning to our study findings, we believe that this may be the first study using the same cohort population to evaluate the association between hypo- and hyperthyroid function with subsequent AMD risk. We found that patients with both hypo- and hyperthyroidism are at much higher risk of developing subsequent AMD than euthyroid persons, levels of odds ratio that have not been reported. Our study may support that physicians should be alerted to this risk in managing patients with thyroid disorders. Apart from the major efforts being made to prevent the development of thyroid diseases, it may be more productive to have an aggressive retinal checkup follow-up and management strategy among patients with thyroid dysfunction as a whole. Preventive actions could include more frequent retinal examinations and dietary antioxidants to prevent AMD, such as higher doses of beta-carotene, vitamins C and E, zinc, lutein, and zeaxanthin, as these supplements are reported to reduce the risk of AMD [27,28,29,30].

This study has several limitations. First, similar to many studies based on health insurance claims data, there remains the possibility of surveillance bias. It is not uncommon for patients with hyperthyroidism to be aware of the possibility of exophthalmos after receiving the diagnosis. While ophthalmopathy is reported to occur in 25–50% of patients with Graves’ disease and 2% of patients with Hashimoto’s thyroiditis, among them, about 3–5% of these patients will develop severe ophthalmopathy, which requires further management [31] Patients with hyperthyroidism may be more likely to frequent outpatient clinics, which may lead to an increased chance of AMD detection due to the increased exposure to medical services. However, such surveillance bias may not be a significant contributor to our finding, as most patients with hyperthyroidism maintain normal vision throughout the disease course, and rarely is a routine ophthalmology clinic visit necessitated by vision symptoms. Also, because patients with hypothyroidism are not expected to experience exophthalmos, they are unlikely to be referred for an eye checkup due to thyroid status. 

A second limitation is that the etiology of AMD remains multifactorial and yet to be clearly documented. Therefore it is not possible to adjust for the possible factors driving AMD incidence in the analysis. We could only include common medical comorbidities for adjustment. There may be some underlying predisposing factors to AMD independent of thyroid function that are not documented to date and not adjusted for in this study.

Third, given that the study is based on claims data, the severity of hyper/hypothyroidism and AMD cannot be captured. Therefore, we were unable to study dose–response relationships, and whether patients with more severe thyroid disorders have greater risk of AMD than those with less severe disease. Despite the limitations of claims-based data, the high numbers of cases and controls studied lends strength to the validity of findings. 

Finally, another limitation of claims data-based study is the absence of data on the extent of treatment of the thyroid disorder, well treated or not. It is possible that at least some of these patients have been well managed and were no longer of dysthyroid status. Most patients with thyroid disorder, if properly managed, return to euthyroid status. With current medical practice norms, it is unlikely that these patients are allowed to remain in a state of poor disease control without regular follow-up. This potential source of bias may have little impact on our conclusions. If too many well-treated subjects were included in the experimental group, theoretically the findings would be biased toward the null hypothesis. Given the effect sizes observed, this limitation may not seriously impact the validity of our conclusions.

Despite the high effect sizes observed in our study, many issues remain to be resolved in using these findings in clinical practice. Due to absence of data on disease severity and status of thyroid disease control, it remains unclear whether proper management of the thyroid disorder reduces the chances of later development of AMD. Further studies are needed to clarify the relationship between these conditions, as to whether it is a causal relationship between the diseases or a consequence of treatment. As disordered thyroid patients were usually followed regularly, it might be possible to conduct a prospective cohort study regarding this issue with all the disease severity and medication dosage recorded and compared in detail. Besides clinical studies, molecular level investigations should also be further explored, especially regarding gene interactions, to provide helpful information in the era of precision medicine. Nevertheless, the high odds of AMD observed in our study, especially for those with hyperthyroidism, suggests the need for precautionary measures against AMD due to the substantial implications for poor quality of life. 

In conclusion, our study shows that patients with thyroid dysfunction are at higher risk of a subsequent diagnosis of AMD. These patients may benefit from aggressive measures to prevent or detect early AMD through a regimen of regular eye examinations regardless of vision status.

## Figures and Tables

**Table 1 jpm-12-00602-t001:** Demographic characteristics of persons with neovascular age-related macular degeneration (AMD) and control patients (*n* = 15,044).

Variable	Patients with Neovascular AMD (*n* = 7522)		Controls (*n* = 7522)		*p* Value
	Total No.	%	Total No.	%	
Age, mean (SD)	68.3 (11.2)		68.3 (12.9)		0.714
Males	4470	59.4%	4454	59.2%	0.791
Monthly Income					0.292
<NT$1~15,841	2369	31.5%	2325	30.9%	
NT$15,841~25,000	2654	35.3%	2607	34.7%	
≥NT$25,001	2499	33.2%	2590	34.4%	
Geographic region					0.921
Northern	3701	49.2%	3740	49.7%	
Central	1826	24.3%	1802	24.0%	
Southern	1822	24.2%	1813	24.1%	
Eastern	173	2.3%	167	2.2%	
Urbanization level					0.764
1 (most urbanized)	2121	28.2%	2192	29.1%	
2	2165	28.8%	2146	28.5%	
3	1253	16.7%	1219	16.2%	
4	1016	13.5%	1000	13.3%	
5 (least urbanized)	967	12.9%	965	12.8%	
Diabetes	2103	28.0%	2111	28.1%	0.885
Hypertension	4138	55.0%	4055	53.9%	0.174
Coronary heart disease	1417	18.8%	1434	19.1%	0.724
Hyperlipidemia	2876	38.2%	2834	37.7%	0.480

**Table 2 jpm-12-00602-t002:** Prevalence and unadjusted odds ratios (ORs) and 95% confidence intervals (CIs) for prior hyperthyroidism or hypothyroidism among patients with neovascular age-related macular degeneration vs. controls.

Presence of Prior Hyperthyroidism	Total (*n* = 15,044)		Patients with Neovascular Age-Related Macular Degeneration (*n* = 7522)		Controls (*n* = 7522)	
	*n*, %		*n*, %		*n*, %	
Yes	99	0.66%	89	1.18%	10	0.13%
No	14945	99.34%	7433	98.82%	7512	99.87%
OR (95% CI)	--		8.995 (4.675–17.307) *p*-value < 0.001		1.000	
Presence of prior hypothyroidism						
Yes	66	0.44%	52	0.69%	14	0.19%
No	14,978	99.56%	7470	99.31%	7508	99.81%
OR (95% CI)	--		3.733 (2.067–6.741) *p*-value < 0.001		1.000	

Notes: OR = odds ratio.

**Table 3 jpm-12-00602-t003:** Covariate-adjusted odds of prior hyperthyroidism (OR and 95% confidence interval, CIs) among neovascular age-related macular degeneration vs. controls (*n* = 15,044).

Variable	Presence of Neovascular Age-Related Macular Degeneration		
	Adjusted OR	95% CI	*p* Value
Prior hyperthyroidism	9.074	4.713–17.471	<0.001
Age	0.999	0.996–1.002	0.594
Sex	1.023	0.957–1.092	0.505
Monthly income			
<NT$15,841 (reference group)	1.0	-	
NT$15,841~25,000	0.992	0.913–1.077	0.848
≥NT$25,001	0.943	0.869–1.024	0.164
Geographic region			
Northern (reference group)	1.0	-	
Central	1.004	0.921–1.096	0.921
Southern	0.999	0.918–1.087	0.979
Eastern	1.012	0.809–1.266	0.917
Urbanization level			
1 (reference group)	1.0	-	
2	1.039	0.953–1.133	0.387
3	1.055	0.951–1.171	0.313
4	1.040	0.927–1.168	0.502
5	1.020	0.907–1.147	0.737
Hyperlipidemia	1.016	0.943–1.094	0.682
Diabetes	0.970	0.897–1.048	0.436
Hypertension	1.050	0.979–1.125	0.170
Coronary heart disease	0.965	0.887–1.050	0.406

**Table 4 jpm-12-00602-t004:** Covariate-adjusted odds of prior hypothyroidism (OR and 95% confidence interval, CIs) among neovascular age-related macular degeneration vs. controls (*n* = 15,044).

Variable	Presence of Neovascular Age-Related Macular Degeneration		
	Adjusted OR	95% CI	*p* Value
Prior hypothyroidism	3.794	2.099–6.858	<0.001
Age	0.999	0.996–1.002	0.398
Sex	1.014	0.950–1.083	0.671
Monthly income			
<NT$15,841 (reference group)	1.0	-	
NT$15,841~25,000	0.993	0.915–1.079	0.872
≥NT$25,001	0.942	0.868–1.023	0.156
Geographic region			
Northern (reference group)	1.0	-	
Central	1.007	0.923–1.098	0.884
Southern	0.998	0.917–1.087	0.970
Eastern	1.022	0.818–1.277	0.850
Urbanization level			
1 (reference group)	1.0	-	
2	1.039	0.953–1.133	0.387
3	1.061	0.956–1.177	0.264
4	1.039	0.925–1.166	0.519
5	1.023	0.910–1.150	0.700
Hyperlipidemia	1.015	0.943–1.093	0.695
Diabetes	0.973	0.990–1.051	0.485
Hypertension	1.047	0.977–1.122	0.193
Coronary heart disease	0.965	0.888–1.050	0.408

## Data Availability

Data from the National Health Insurance Research Database, now managed by the Health and Welfare Data Science Center (HWDC), can be obtained by interested researchers through a formal application process addressed to the HWDC, Department of Statistics, Ministry of Health and Welfare, Taiwan (https://dep.mohw.gov.tw/DOS/lp-2506-113.html, accessed on 2 January 2022).

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
