# Peer review of "Association of Age-Related Macular Degeneration with Prior Hyperthyroidism and Hypothyroidism: A Case–Control Study"

_jpm, 2022, doi:10.3390/jpm12040602_

Round 1

Reviewer 1 Report

In this study by Hung et al., the authors have looked at the association between age related macular degeneration (AMD) and hypo- or hyper-thyroidism. The authors have looked into Taiwan’s national database for >7000 patients older than 40 years and reported to suffer from AMD. They find there are significantly higher chances of developing AMD in patients with a prior history of hypothyroidism and hyperthyroidism. The study does an exhaustive correlation analysis and confirm that there is a positive association between AMD and the above mentioned thyroid diseases. The authors also discuss the limitations of the study and future studies could be focused on addressing those specific issues.

Minor comments- 

1) On page 18, second paragraph, please correct the reference for Ma et. al.

2) The authors should comment on the possibility of co-existing ophthalmic diseases such as diabetic retinopathy which could be highly prevalent in patients with thyroid related problems.

Author Response

1)On page 18, second paragraph, please correct the reference for Ma et. al.

Response: We have corrected the reference accordingly.

2) The authors should comment on the possibility of co-existing ophthalmic diseases such as diabetic retinopathy which could be highly prevalent in patients with thyroid related problems.

Response: Thank you. We have added these comments accordingly as follows:

“Another possible explanation is through the influence of diabetes mellitus that as patients with diabetes have a higher incidence of thyroid disease than people without, the coexistence of diabetes with thyroid disease leads to retinal endothelial damage. The degree of retinal endotherlial cell dysfunctions will significantly impact the course of macro-and microangiopathic complications in these retinopathy patients [26].”(page 21)

Reviewer 2 Report

REVIEWER’S COMMENTS

The manuscript Association of Age-related Macular Degeneration with Prior Hyperthyroidism and Hypothyroidism: A Case-Control Studyby Hung et al shows that patients with thyroid dysfunction are at higher risk of a subsequent diagnosis of AMD.

  1. Please present the Mean ± SEM values as well.
  2. Please discuss briefly the future directions for this research.
  3. Please proofread for grammatical, spelling, and syntax errors throughout the paper
  4. Please be consistent with the style of references.

Author Response

1)Please present the Mean ± SEM values as well.

Response: We have presented mean ± standard deviation in Table 1. (page 12)

2)Please discuss briefly the future directions for this research.

Response: Thank you. We have discussed briefly accordingly as follows:

“As disordered thyroid patients were usually followed regularly, it might be possible to conduct a prospective cohort study regarding this issue with all the disease severity and medication dosage recorded and compared in detail. Besides clinical studies, molecular level investigations should also be further explored, especially regarding gene interactions, to provide helpful information in the era of precision medicine.”(page 24)

3)Please proofread for grammatical, spelling, and syntax errors throughout the paper

Response: The co-author Sudha Xirasagar, a native English speaker, has proofread this manuscript.

4)Please be consistent with the style of references.

Response: We have revised them accordingly.